# Self-Completion Questionnaire on Sleep Evaluation in Patients Undergoing Oxaliplatin Therapy: An Observational Study

**DOI:** 10.3390/cancers16050946

**Published:** 2024-02-26

**Authors:** Maria Valentina Mussa, Sarah Allegra, Tiziana Armando, Silvana Storto, Beatrice Ghezzo, Giulia Soave, Giuliana Abbadessa, Francesco Chiara, Massimo Di Maio, Fiammetta Maria Dagnoni, Silvia De Francia

**Affiliations:** 1Centro Oncoematologico Subalpino (COES), Molinette University Hospital of the City of Health and Science of Turin, 10126 Turin, Italy; mariavalentina.mussa@unito.it (M.V.M.); tarmando@cittadellasalute.to.it (T.A.); silvana.storto@unito.it (S.S.); soavegiu@gmail.com (G.S.); massimo.dimaio@unito.it (M.D.M.); fiammetta.dagnoni@gmail.com (F.M.D.); 2Laboratory of Clinical Pharmacology “Franco Ghezzo”, Department of Clinical and Biological Sciences, University of Turin, S. Luigi Gonzaga Hospital, 10043 Orbassano, Italy; giuliana.abbadessa@unito.it (G.A.); 336124@edu.unito.it (F.C.); silvia.defrancia@unito.it (S.D.F.); 3Specialization School in Ophthalmology, Department of Surgical Sciences, University of Turin, 10124 Turin, Italy; ghezzob@gmail.com

**Keywords:** sleep complaints, Pittsburgh Sleep Quality Index, oxaliplatin, cancer, health, life quality

## Abstract

**Simple Summary:**

Sleep is a fundamental human need. Good-quality sleep has a protective role that allows people to carry out daily activities adequately, ensuring well-being. Prolonged wakefulness causes a wide spectrum of disabilities. This study aims to understand the effects of sleep complaints on health and quality of life for cancer patients, and it has been conducted through the creation of questionnaires for patients to investigate the potential alteration of the sleep pattern. The results obtained may serve as the foundation for designing awareness campaigns, providing information, and updating patients, healthcare personnel, and caregivers. It is definitively important, in fact, to begin an evaluation of the negative effects of sleep complaints in the field of cancer. The model developed can also be replicated in different clinical settings: data adjustment by different parameters (i.e., geographic locations, age, sex, and gender) can be the key to improving the patient’s quality of life.

**Abstract:**

Sleep is a fundamental human need; sleep disruption, in fact, causes an increase in the activity of the sympathetic nervous system and the hypothalamic-pituitary-adrenal axis, metabolic effects, changes in circadian rhythms, and pro-inflammatory responses. The scientific literature is finally starting to pay attention to the central role of sleep alterations in patients health. Oxaliplatin is extensively used for the treatment of gastrointestinal cancer and other malignancies, with an increased frequency of use in recent years. This study aims to understand the effects of sleep complaints on health and quality of life in cancer patients treated with oxaliplatin. A study has been conducted through the creation and distribution of questionnaires to patients to investigate their complaints about sleep quality. We observed significant differences between males and females in evaluating sleep hygiene scores, the Pittsburgh Sleep Quality Index, and previous difficulty sleeping. Moreover, in females, stress, worries, and anxiety seem to play a negative role in the sleep hygiene score. The obtained results could improve the interest of healthcare personnel and caregivers in sleep quality in patients undergoing chemotherapy.

## 1. Introduction

Sleep is a fundamental human need. Good-quality sleep has a protective and restorative function that allows people to carry out their daily activities in an adequate way, ensuring physical, social, and emotional well-being. Prolonged wakefulness causes a wide spectrum of harmful physiological, psychological, and behavioral effects. Sleep alteration, in fact, causes an increase in the activity of the sympathetic nervous system and the hypothalamic-pituitary adrenal axis, metabolic effects, changes in circadian rhythms, and pro-inflammatory responses. According to scientific research, people with cancer are more likely than the general population to experience sleep disruptions, which result in very fragmented sleep with a preference for the shallower, less restorative phases of sleep. Many different effects are due to sleep–wake rhythm alterations. Increased stress reactivity, somatic pain, a decline in quality of life, emotional and mood disorders, cognitive deficiencies, memory, performance, and delirium are only a few of the short-term effects. Hypertension, dyslipidemia, cardio-vascular disease, issues with weight, metabolic syndrome, type 2 diabetes, and colorectal cancer are long-term effects [1,2,3]. Over 50% of cancer patients suffer from insomnia, nearly twice the estimated prevalence in the general population. Cancer and its treatments disturb sleep–wake functioning; however, there are not yet many studies about sleep complaints in cancer patients [4,5,6,7,8]. The evaluation of the detrimental effects of a change in sleep–wake rhythm is then crucial in oncological pain [9]. Sleep disorders are prevalent in patients with advanced cancer, as indicated in a paper by Innominato and colleagues [10]. Bernatchez and colleagues conducted a study in palliative cancer patients aimed at characterizing patients’ sleep–wake cycles using various circadian parameters, showing how the sleep–wake cycles are markedly disrupted in palliative cancer patients, especially near the end of life [11]. Berger and colleagues also showed that fatigue is the most prevalent and distressing symptom experienced by patients receiving adjuvant chemotherapy for early-stage breast cancer, and higher fatigue levels have been related to sleep maintenance problems [12]. Oxaliplatin is a commonly used platinum-based chemotherapy drug for colorectal cancer and other malignancies, with an increased frequency of use in recent years. A strong relationship between cancer and disrupted sleep–wake patterns has already been widely investigated in the literature generally, but very little evidence is currently present related to cancer patients treated with oxaliplatin. Yoshikawa and colleagues conducted a study to investigate quality of life and nighttime sleep disturbance in colon cancer patients treated with oxaliplatin-based chemotherapy [13]. Considering cardiotoxicity, Predisposing factors for cardiotoxicity, the main adverse effect due to oxaliplatin high dosage, should also be sleep disturbance due to obstructive sleep apnea, as well as age over 60 years old, valvular heart disease, hypertension, chronic renal disease, diabetes mellitus, and smoking, which are common risk factors for atrial fibrillation [14]. Moreover, Zheng and colleagues suggested that using natural antioxidants like melatonin, mainly used to treat sleep disorders and jetlag, should be the standard clinical strategy to reduce the cardiotoxicity caused by anticancer drugs. By scavenging reactive oxygen species (ROS) and reactive nitrogen species (RNS), melatonin metabolites also demonstrate antioxidant activity [15]. In addition, it has recently been demonstrated that Nod-like receptor family pyrin domain-containing 3 (NLRP3) inflammasome activation is a critical mechanism in sleep modulation [16]. NLRP3 also controls the release of pro-inflammatory cytokines IL-1β/IL-18 and chemokines linked to the proliferation of cancer cells and heart damage [17]. Cardiotoxicity brought on by various anticancer medications and the advancement of cardiometabolic disorders are both significantly influenced by cytokines in cardiac tissue.

According to recent research, immune checkpoint drugs and chemotherapy for cancer patients also worsen cardiac damage caused by interleukin1-β. As discussed by Quagliarello et al., cancer patients are more likely than the general population to experience cardiovascular issues, particularly in the era of the SARS-CoV-2 pandemic, which has been linked to coagulopathies, myocarditis, and heart failure [18].

Here, patients completed a questionnaire about chemotherapy-induced nausea and vomiting and sleep disturbance. Sleep disturbance was checked, and the relationship between sleep disturbance and nausea was analyzed, leading to the result that nausea affects the quality of life and nighttime sleep of colon cancer patients. To conclude, even less evidence is present in the literature focusing on sleep alterations in cancer patients treated with oxaliplatin, considering sex disaggregated data. Women experience sleep differently from men: each phase of a woman’s life, from childhood to menopause, increases the risk of sleep disturbance [19]. It is clearly agreed that sleep disorders represent both a cancer risk factor and a biological consequence of the activation of the immuno-inflammatory system induced by cancer itself. As a consequence, it is clear from the literature that sleep disorders in cancer are then different among sexes and more common in women [20]. Further studies are evidently needed.

## 2. Materials and Methods

### 2.1. Patients and Inclusion Criteria

This was a descriptive retrospective study carried out at the Oncology Day Hospital Unit (COES) of the Molinette Hospital AOU Città della Salute e della Scienza in Turin (Italy) between May 2023 and May 2024. The inclusion criteria were as follows: age ≥ 18 years old, ability to understand and agree to this study, Italian language, and actively undergoing chemotherapy with oral or intravenous formulations of oxaliplatin. The exclusion criteria were as follows: age under 18 years old, assumptions about other chemotherapy agents. No data were collected regarding the start of therapy or the ongoing chemotherapy cycle.

### 2.2. Self-Completion Questionnaire

A self-completion questionnaire has been built to achieve the goal of our study. The first part of the questionnaire consisted of patient demographic data, including age, gender, highest level of education, marital status, tumor localization (lung, intestine, esophagus-gut, liver and biliary tract, pancreas and genitourinary tract), oral or intravenous oxaliplatin treatment, central venous catheter, stoma, concomitant therapies (corticosteroids, antiemetics, antipyretics, treatment of cardiovascular disease, insulin and oral hypoglycemic agents, antidepressant, treatment of thyroid disorders, antihistaminic and antipsychotic), concomitant pathology (cardiovascular diseases, diabetes, endocrine disorders, psychiatric disorders, arthritis/arthrosis, osteoporosis and rheumatic diseases, gastric disorders, intestinal diseases, genitourinary diseases, neurological diseases, respiratory disorders), parenteral nutrition, pain, consumption of drug to treat pain (antipyretics, non-steroidal anti-inflammatory drugs, opioids), previous difficulty sleeping (before starting any chemotherapy; yes/not), stress, worries and anxiety (yes/not) and insomnia and treatment for sleeping difficulties (phytotherapies, melatonin, chamomile, other drugs). The second part consists in the administration of a questionnaire relating to sleep hygiene [21], aimed at understanding and knowing patients main habits relating to their sleep through the evaluation of 22 questions, considering how many times a week in the last month have been implemented. In order to calculate the sleep hygiene score, it is necessary to add the scores obtained. To interpret the results, it must be considered that higher scores indicate poorer sleep hygiene. The last part consists of the Pittsburgh Sleep Quality Index (PSQI) [22], which is the most used questionnaire to detect sleep disorders and the global quality of self-sleep—reported in clinical or non-clinical contexts, considering the month prior to completion—through 19 items that investigate seven domains: subjective sleep quality, sleep latency, sleep duration, habitual sleep efficiency, sleep disturbances, use of sleep medications, and daytime dysfunction. For this questionnaire, we used the version validated in Italian in 2013 [23]: the authors state that the questionnaire’s validation maintains a high degree of coherence, that it correctly discriminates between patients suffering from pathologies that worsen the quality of sleep and healthy participants belonging to the control group, both young and older, as well as confirming that the score used to differentiate the population with good sleep quality from the bad one (the threshold value indicating good sleep quality is five; above this number, sleep quality is worst) is also applicable to the Italian population. Therefore, in summary, we obtained a single score from the Sleep Hygiene questionnaire, a PSQI score, and a PSQI greater or less than five.

The questionnaire has been distributed to the patients included in this study. In order to achieve the set objectives, a database was built in which the data were codified and collected.

Each patient provided written informed consent to fill out the questionnaire, and we received approval for this study from the hospital directors. Participation was voluntary, anonymous, and without compensation; therefore, each patient was free to choose whether to complete the questionnaire or not.

These data were routinely recorded during daily clinical practice as a quality assurance measure and to explore improvements in the quality of services; therefore, ethics committee approval was not required. Confidentiality was guaranteed in the data collection, analysis, and dissemination phases, presenting the results in aggregate form.

### 2.3. Statistical Analysis

All the variables were tested for normality with the Shapiro–Wilk test. The correspondence of each parameter was evaluated with a normal or non-normal distribution through the Kolmogorov–Smirnov test. Quantitative variables were described via median and interquartile range (IQR, quartile 1; quartile 3) if the variables were not normally distributed. Qualitative variables were described via frequencies and percentages. The data were analyzed separately for male and female patients. Kruskal–Wallis and Mann–Whitney tests were used to evaluate the influence of variables on the sleep hygiene score, PSQI score, and PSQI cutoffs (lower than 5 and higher than 5), considering the level of statistical significance (*p* < 0.05).

Any predictive power of all the considered variables was finally evaluated through univariate and multivariate linear (for sleep hygiene score and PSQI) and logistic (considering PSQI above 5) regression analyses. Factors [β, β coefficient for linear model; Exp(B), exponentiation of the B coefficient for logistic model; confidence interval (CI) at 95% for the parameter B, for linear model, or Exp(B) for logistic model] with a *p* value less than 0.2 in univariate analysis were included in the multivariate analysis (*p* < 0.05).

All the tests were performed with IBM SPSS statistics 25.0 for Windows (SPSS Inc., Chicago, IL, USA).

## 3. Results

### 3.1. Study Population

A total of 144 patients were enrolled; 42.4% were females and 57.6% were males; their demographic characteristics, sleep hygiene score, and PSQI statistic results are shown in Table 1.

The distribution of their social, clinical, and therapeutic characteristics is resumed in Table 2.

### 3.2. Influence of Sex on Sleep Quality

Evaluating the influence of sex on sleep quality, we observed statistically significant differences regarding sleep hygiene score, with a female median value of 1.68 (IQR 1.32–2) versus 1.36 (IQR 1.14–1.82) for males (Figure 1).

A borderline influence has also been observed when evaluating the PSQI (*p =* 0.057), with a female median value of 6 (IQR 4–10) and a male median value of 5 (IQR 4–8). Instead, regarding the PSQI score below or above 5, a statistically significant difference has been observed comparing females and males (*p =* 0.013; Figure 2).

After evaluating the previously reported sleeping difficulties, we observed a statistically significant difference in distribution between sexes (*p* = 0.017; Figure 3).

Evaluating men and women separately, we reported that in females, the declared previous difficulty sleeping, thus before the starting of chemotherapy, negatively influenced PSQI, with a median value of 10 (IQR 7–13) versus the 6 (IQR 3.75–9) of those without previous difficulty (*p* = 0.004) (Figure 4).

Again, in female patients, reported stress, worries, and anxiety influenced sleep hygiene scores (*p* = 0.048), with a score of 1.68 (IQR 1.31–1.95) versus 2.2 (IQR 1.9–2.57) in those without these symptoms (Figure 5).

### 3.3. Regression Analyses

Considering the sleep hygiene score, no variables were included in the linear regression model. Evaluating the PSQI, we observed a predictive role of phytotherapic use (*p* = 0.04; β = −0.490; IC95% −14.987 and −0.413); in addition, a borderline role resulted considering the sex variable (*p* = 0.053; β = −0.457; IC95% −7.121 and 0.055).

Carrying out a logistic model to evaluate the predictive role of collected variables on PSQI score above 5, we reported that only sex has been retained with a *p* = 0.001 (Exp(B) = 0.139; IC95% 0.042 and 0.462).

## 4. Discussion

From the analysis of the collected data and in accordance with the literature [13,23,24,25,26,27,28,29], it was possible to identify the presence of sleep complaints in patients undergoing oncological treatment in our study of oxaliplatin-based therapy who participated in completing the questionnaire. Thus, we can see, through the results of the PSQI questionnaire, that more than 77% of the participants have a poor quality of sleep, and this is observed above all in females. There are studies in the literature that use the same questionnaire to investigate sleep quality in cancer patients, whether they are undergoing treatment or not. In a 2020 observational study, despite the limited sample of patients examined (92 participants), results similar to our research were found: 57.6% showed a PSQI score > 5 and an average of 6, 57 which indicates poor sleep quality [30]. Even in a 2022 meta-analysis, sleeping disorders were found in 64% of patients using the PSQI questionnaire [31]. And again, in a 2022 review, the authors compare the mean PSQI scores before, during, and after oncological treatment with the following results: 7.11–8.31–7.10 (all statistically significant for *p* < 0.001), thus demonstrating that patients undergoing treatment suffer more from sleep alterations with a lower quality of sleep [23]. Considering our observed sex differences, studies in the literature report that sleep disorders within the general population are more frequent in females [32]. Regarding cancer patients undergoing oncological treatment, many studies showed that female sex is significantly correlated with lower sleep quality [25,31,33,34,35,36]. Within a 2022 meta-analysis study, the association between sleep disorders and female sex was found to be statistically significant, mainly due to the lower possibility of having social support and the higher exposure and affection caused by pathologies such as anxiety and depression [31]. Conversely, in a 2021 study in which the sleep quality of patients was investigated through the compilation of numerous validated questionnaires, the correlation between PSQI scores and female gender was not statistically significant [37]. We observed a trend towards better sleep quality in women, with a higher median value of the sleep quality index. But, in our population, there are approximately twice as many men with a PSQI below five, thus with good sleep quality, compared to female patients; in addition, linear and regression analyses on PSQI and PSQI above five showed that male sex has a predictive role for better sleep.

In addition to investigating the quality of sleep during the treatment, the patients were asked if they had suffered from sleep complaints or insomnia before the start of oncological therapy. Some studies in the literature show an investigation into the quality of the patient’s sleep through the compilation of the PSQI questionnaire in at least two different periods, i.e., before and during treatment, both to evaluate the effect of therapy and to take into consideration previous problems [23,38]. From the results obtained, there is a statistically significant correlation between previous difficulty sleeping and the quality of sleep during oncological treatment (*p* < 0.001) in 20.9% of participants, showing an average score higher than those who report the absence of sleeping disorders before treatment. These data indicate that those with previous problems have a higher risk of having the worst sleep quality during treatment. In our study, in female patients who declared previous difficulty sleeping, a higher PSQI and thus worse quality sleep resulted. Moreover, compared to the male population, over twice as many women reported having difficulty sleeping even before beginning oxaliplatin medication.

In our study, age does not represent a statistically significant variable correlated to the disorders of the sleep quality, contrary to what is reported in the literature, in which advanced age is associated with a significant worsening of the sleep–wake rhythm during oncological therapy, in particular considering intravenous chemotherapy and patients over 75 years old, due to the greater presence of comorbidities [13,27,34,35,39]. In a 2020 review study, however, age significantly affects poor sleep quality in both older and younger patients [25]. Conversely, in other works, it emerges that older patients experience sleep disturbances to a lesser or similar extent compared to young people [40], but the greatest prevalence of disorders is found in the age group between 40 and 49 years old [31].

Both research and scientific evidence showed that there is no statistically significant correlation between PSQI score and marital status [37].

In the data collection of our study as well as in the literature, the various primary oncological pathologies of the selected sample were examined [23,34,39,41,42]. In the selected scientific studies, it is claimed that based on the type of tumor, a different load of symptoms depends, as well as a variation in the quality of sleep, in particular in patients with lung cancer due to the presence of respiratory problems [41,43,44]. Unfortunately, the great variety of oncological pathologies and the relatively small sample size of our study did not allow us to carry out an effective statistical analysis.

The associations between sleep alterations, type of treatment, and chemotherapy drugs do not provide statistically significant data. The relatively small sample size may also have influenced this result, especially for some variables with lower frequencies. In the literature, some studies agree with the same result found [30,43], while within a 2022 meta-analysis, the authors argue that there are higher levels of sleep disorders in patients subjected to a mixed treatment, and this is also given by treatment-related toxicities [31].

According to the literature, a greater presence of comorbidities is correlated with an equally deterioration of the sleep–wake rhythm [39]; in particular, anxiety and depression are relevant and have an impact on sleep [37,45,46,47,48,49]. We observed that self-reported stress, worries, and anxiety negatively influenced the sleep quality score, with a better quality of sleep in those who do not experience these symptoms.

In our cohort, we have not observed significant influences of the use of concomitant drugs, even those used to treat insomnia, on the quality of sleep. Conversely, in the literature, corticosteroids, antiemetics, and antidepressants are reported as drugs that worsen sleep quality, particularly by altering sleep maintenance and latency [37]. Another drug category that could affect the sleep–wake rhythm is represented by opioids, since they cause nocturnal breathing problems and a state of sedation during the day, as well as anxiety and insomnia [37,50,51]. According to some evidence, however, these medicines could also have positive effects by improving the quality of sleep; for example, the regular administration of low quantities of morphine can reduce respiratory problems [52].

Contrary to what we observed, the evidence present in the literature states that sleep quality and pain are negatively correlated [27,35,51], such that sleep disorder symptoms could cause pain threshold reduction [53]. Even according to a 2020 review, pain and sleep are physiological mechanisms connected to each other; in fact, severe pain is associated with a worse quality of sleep [25].

Eventually, we observed the PSQI predictive role of phytotherapy, therefore hypothesizing that it can have positive effects on the quality of sleep. The general population finds non-pharmacological therapies especially appealing since they are thought to have no negative side effects [54]. Among the most popular complementary and alternative medicine choices are herbal medicines and medicinal herbs [55,56,57]. The general public especially welcomes them for sleep disorders. Researchers worldwide have assessed and shown the greatest interest in the following plants for sleep disturbances: *Valeriana officinalis*), *Lavandula angustifolia*, *Humulus lupulus*, *Matricaria chamomilla*, *Crataegus monogyna*, *Hypericum perforatum*, and *Rosmarinus officinalis* [58].

The importance of sleep education for cancer patients during chemotherapy periods to improve the quality of sleep has already been highlighted [59]. Within this research, a score was obtained through the relevant questionnaire used to quantify the patient sleep hygiene, with a median PSQI of 6 (IQR 4–9), which remains almost unchanged in males; in female patients instead, the data get worse with a medina value of 5 (IQR 4–8).

Good-quality sleep can improve energy, quality of life, and coping, refers to the attitudes and behaviors necessary to maintain emotional well-being and to adjust the diagnosis-caused stress in cancer patients. The literature underlines the importance of the nursing figure and of the knowledge of phases of sleep, its disorders, and the main factors that can cause them [60]. Promotion and treatment interventions can be implemented, as well as the integration of a specific sleep assessment into ordinary care in a multidisciplinary approach [25,35]. According to a 2022 review, it is important for the nurse to have an active role in this process, collaborating with the patients themselves and their families to choose the types of interventions to counteract sleep–wake rhythm disorders and to carry out the related monitoring [61]. The role of the nurse could be upgraded, both through research, conducting more specific studies and with larger samples that can better reflect the sleep quality of cancer patients undergoing systemic treatment, and by raising awareness among staff to introduce assessments into daily clinical practice.

This study has limitations. The first of all is the data collection method: using a “self-reported” questionnaire filled out freely by the patient without medical consultation; for this reason, some information may be inaccurate or incomplete. Furthermore, no data were collected regarding the start of therapy or the ongoing chemotherapy cycle; the patient was asked to complete the questionnaire during the treatment period without considering the time that had passed since its start or the number of sessions already carried out. A more precise study could be produced by examining a predefined sample of patients and administering the questionnaire to them more than once at predefined times during therapy to evaluate the changes in sleep quality. In addition, in this research, there are no variables that define the staging of the tumor or the presence of metastases; both are aspects that, according to evidence, can influence the quality of sleep [27,35]. Evaluating two other studies, however, it has been demonstrated that there is no statistically significant association between PSQI score and tumor staging, while an association is found between alteration of the sleep–wake rhythm and metastases, in particular visceral and bone ones [37,43].

An aspect that was not investigated in this research but was considered in other studies was the influence of physical activity on sleep in cancer patients. In a longitudinal study, it was highlighted how patients who do not perform physical activity presented higher levels of fatigue in the morning, sleep disturbances, and lower levels of energy in the morning and evening compared to those who carry it out [62]. Moreover, in a 2019 meta-analysis, the importance for cancer patients of maintaining physical activity and exercise practice, especially the aerobic type, could improve their quality of sleep [63].

## 5. Conclusions

In conclusion, from this research, it emerges that patients undergoing oncological therapies present overall poor sleep quality, and that these disorders are associated with anxiety, worries, depression, and previous difficulty sleeping in females, and tumor localization in males.

## Figures and Tables

**Figure 1 cancers-16-00946-f001:**
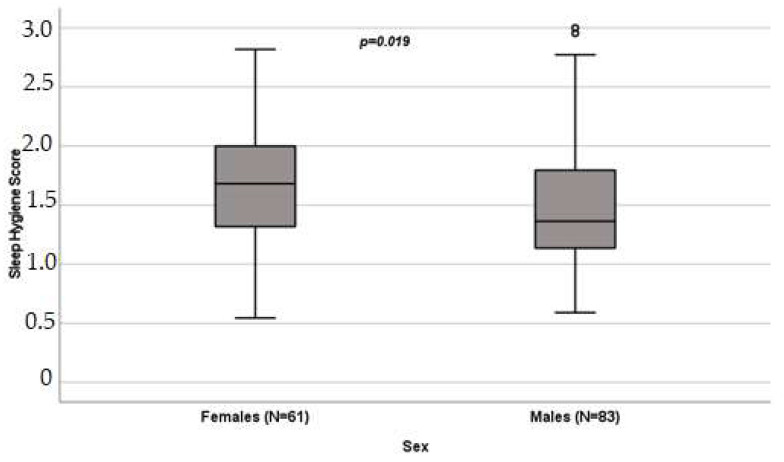
Influence of sex on sleep hygiene score (*p* = 0.019) in patients undergoing oxaliplatin chemotherapy, obtained with the Mann–Whitney test. Box plot of sleep hygiene score distribution in female and male patients; boxes and black lines in boxes represent, respectively, interquartile ranges (IQRs) and median values; open dots and stars represent outlier values. Median values (horizontal line), IQR (bars), patient values (black square), highest and lowest values (whiskers), and *p* value are shown.

**Figure 2 cancers-16-00946-f002:**
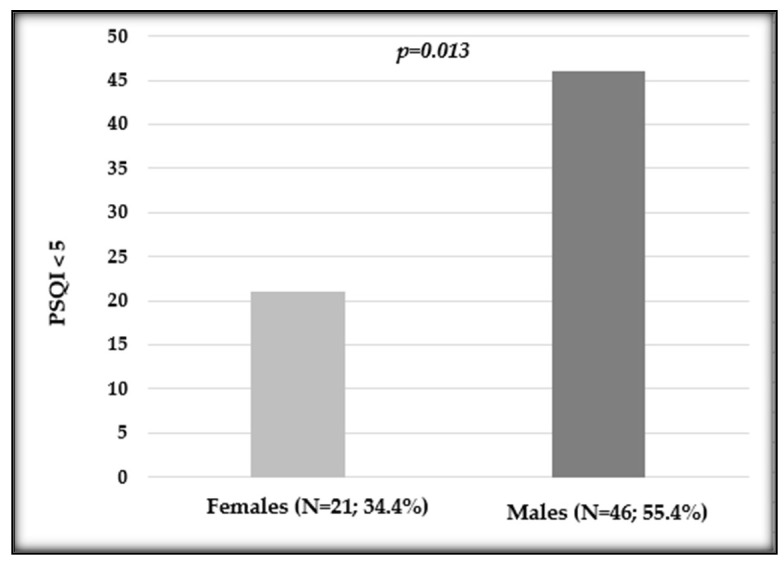
Distribution of the Pittsburgh Sleep Quality Index (PSQI) in female and male patients undergoing oxaliplatin chemotherapy. Number and percentage of patients in each variable and *p* value are shown.

**Figure 3 cancers-16-00946-f003:**
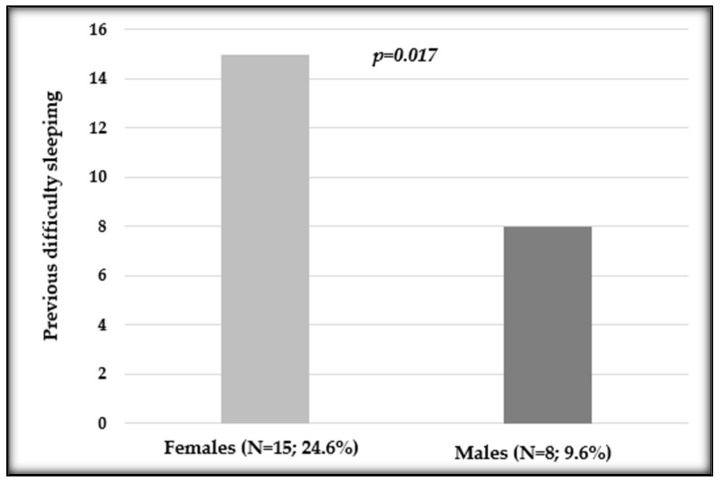
Distribution of previous difficulty sleeping in female and male patients undergoing oxaliplatin chemotherapy. Number and percentage of patients in each variable and *p* value are shown.

**Figure 4 cancers-16-00946-f004:**
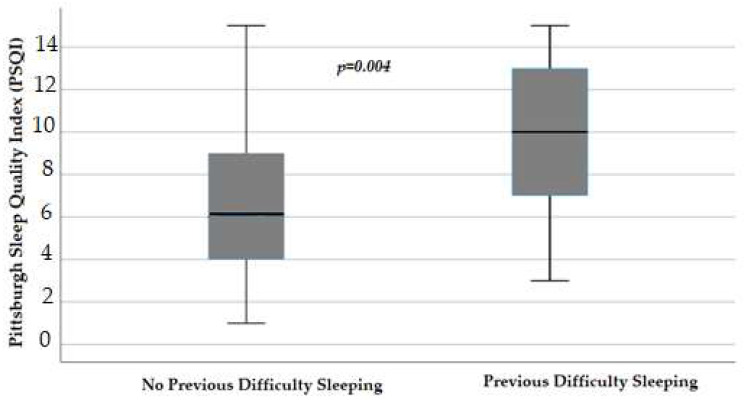
Influence of difficulty sleeping before the starting of chemotherapy on the Pittsburgh Sleep Quality Index (*p* = 0.004) in female patients, obtained with Mann–Whitney test. Box plot of sleep Pittsburgh Sleep Quality Index distribution in female patients; boxes and black lines in boxes represent respectively interquartile ranges (IQRs) and median values; open dots and stars represent outlier values. Median values (horizontal line), IQR (bars), patient values (black square), highest and lowest values (whiskers), and *p* value are shown.

**Figure 5 cancers-16-00946-f005:**
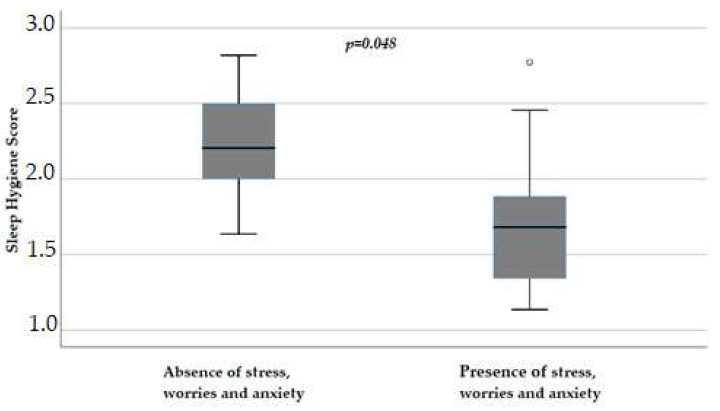
Influence of stress, worries, and anxiety on sleep hygiene score (*p* = 0.048) in female patients, obtained with Mann–Whitney test. Box plot of sleep hygiene score distribution in female patients; boxes and black lines in boxes represent, respectively, interquartile ranges (IQRs) and median values; open dots and stars represent outlier values. Median values (horizontal line), IQR (bars), patient values (black square), highest and lowest values (whiskers), and *p* value are shown.

**Table 1 cancers-16-00946-t001:** Demographical characteristics, sleep hygiene score, and PSQI statistic results.

Variable	All (*n* = 144)	Males (*n* = 83)	Females (*n* = 61)
Age (years)			
Median (IQR)	62 (54–68)	62 (56–68)	59 (52–67.5)
Sleep Hygiene score			
Median (IQR)	1.5 (1.18–1.90)	1.36 (1.13–1.81)	1.68 (1.31–2)
Pittsburgh Sleep Quality Index (PSQI)			
Median (IQR)	6 (4–9)	5 (4–8)	6 (4–10)
PSQI ≤ 5 (percentage, frequency)	67, 46.5%	46, 55.4%	21, 34.4%
PSQI > 5 (percentage, frequency)	77, 53.5%	37, 44.6%	40, 65.6%

**Table 2 cancers-16-00946-t002:** Distribution of social, clinical, and therapeutic characteristics of the enrolled patients.

Variable	All (*n* = 144)	Males (*n* = 83)	Females (*n* = 61)
Highest level of education (Frequency, Percentage)			
Primary School Diploma	4, 2.8%	3, 3.6%	1, 1.6%
Middle School Diploma	49, 34%	29, 34.9%	20, 32.8%
High School Diploma	62, 43.1%	36, 43.3%	26, 42.6%
Degree	28, 19.4%	14, 16.9%	14, 23%
Marital Status (Frequency, Percentage)			
Married—Cohabiting	106, 73.6%	64, 77.1%	42, 68.9%
Separated—Divorced	17, 11.8%	10, 12%	7, 11.5%
Unmarried Maiden	12, 8.3%	7, 8.4%	5, 8.3%
Widow	8, 5.6%	1, 1.2%	7, 11.5%
Tumor Localization (Frequency, Percentage)			
Lungs	6, 4.2%	4, 4.8%	2, 3.3%
Intestine	92, 63.9%	51, 61.4%	41, 67.2%
Stomach, Esophagus	27, 18.8%	19, 22.9%	8, 13.1%
Liver And Biliary Tract	5, 3.5%	2, 2.4%	3, 4.9%
Pancreas	13, 9%	8, 9.6%	5, 8.2%
Genitourinary Tract	1, 0.7%	0	1, 1.6%
Chemotherapy Administration (Frequency, Percentage)			
Intravenous	144, 100%	83, 100%	61, 100%
Oral	23, 16%	15, 18.1%	8, 13.1%
Central Venous Catheter (Frequency, Percentage)			
Present	143, 99.1%	82, 98.8%	61, 100%
Stoma (Frequency, Percentage)			
Present	16, 11.1%	11, 13.3%	5, 8.2%
Nutrition (Frequency, Percentage)			
Oral	141, 97.9%	80, 96.4%	59, 96.7%
Parenteral	3, 2.1%	2, 2.4%	1, 1.6%
Pain Sensation (Frequency, Percentage)			
Present	32, 22.2%	15, 18.1%	17, 27.9%
Pain Relievers (Frequency, Percentage)			
Paracetamol	20, 13.9%	10, 12%	10, 6.4%
Non-Steroidal Anti-Inflammatories	4, 2.8%	2, 2.4%	2, 3.3%
Opioids	11, 7.6%	6, 7.2%	5, 8.2%
Concomitant Pathologies (Frequency, Percentage)			
Cardiovascular Diseases	48, 33.3%	27, 32.5%	21, 34.4%
Diabetes	15, 10.4%	12, 14.5%	3, 4.9%
Endocrine Disorders	9, 6.3%	2, 2.4%	7, 11.5%
Psychiatric Disorders	6, 4.2%	2, 2.4%	4, 6.6%
Arthritis, Arthrosis, Osteoporosis, Rheumatic Disorders	4, 2.8%	1, 1.2%	3, 4.9%
Gastric Disorders	3, 2.1%	1, 1.2%	2, 3.3%
Intestinal Disorders	1, 0.7%	1, 1.1%	0
Genitourinary Disorders	2, 1.4%	2, 2.4%	0
Neurological Disorders	3, 2.1%	3, 3.6%	0
Respiratory Disorders	0	0	0
Concomitant Therapies (Frequency, Percentage)			
Corticosteroids	68, 47.2%	32, 38.6%	36, 59%
Antiemetics	67, 46.5%	35, 42.2%	32, 52.5%
Drugs for Cardiovascular Diseases	56, 38.9%	36, 43.4%	20, 32.8%
Antidepressants	7, 4.9%	2, 2.4%	5, 8.2%
Benzodiazepines	7, 4.9%	4, 4.8%	3, 4.9%
Drugs for Thyroid Diseases	9, 6.3%	2, 2.4%	7, 11.5%
Insulin And Oral Hypoglycemics	13, 9%	10, 12%	3, 4.9%
Antihistamines	8, 5.6%	6, 7.2%	2, 3.3%
Antipsychotics/Mood Stabilizers	0	0	0
Antiepileptics/Anticonvulsants	1, 0.7%	1, 1.2%	0
Previous Difficulty Sleeping (Frequency, Percentage)			
Yes	23, 16%	8, 9.6%	15, 24.6%
Treatment for Difficulty Sleeping (Frequency, Percentage)			
Phytotherapeutics	2, 1.4%	1, 1.2%	1, 1.6%
Melatonin	1, 0.7%	0	1, 1.6%
Chamomile and Herbal Teas	4, 2.8%	3, 3.6%	1, 1.6%
Drugs	10, 6.9%	3, 3.6%	7, 11.5%
Stress/Worries/Anxiety (Frequency, Percentage)			
Yes	18, 12.5%	11, 13.25%	7, 11.48%

## Data Availability

Data are contained within the article.

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
