# Peer review of "Self-Completion Questionnaire on Sleep Evaluation in Patients Undergoing Oxaliplatin Therapy: An Observational Study"

_cancers, 2024, doi:10.3390/cancers16050946_

Round 1

Reviewer 1 Report (New Reviewer)

Comments and Suggestions for Authors

Manuscript titled “ Self-completion questionnaire on sleep evaluation in patients undergoing oxaliplatin therapy: an observational study. “ is a very interesting article in the field of QoL in cancer patients. The overall structure is of good quality and easy to read. Methods and Results are clear and results corroborate the initial hypothesis of the authors. Figures and Tables are of sufficient quality and easy to read as well as to understand to readers.

However, manuscript need some improvements, specifically in Introduction and/or Discussion. Here the points:

1. In introduction, authors should describe not only the role of QoL in oxaliplatin-treated patients but also the incidence of cardiotoxicity and its possible correlation with sleep disturbance.

2. No high sleep quality is related to high pro-inflammatory cytokines and NLRP3 inflammasome. Authors should describe this link and highlight on the key role of interleukin 1 beta in cancer survival and cardiovascular diseases ( cite doi: 10.26355/eurrev_202111_27124.)

Based on these changes, the article could be suitable for publication in this journal.

Author Response

Turin, February, 2024

Dear Editor,

Please, find enclosed a revised manuscript (Manuscript ID: cancers-2824919) to be considered for publication in Biomedicines, Special Issue on " Platinum-Based Therapeutics for Cancer” as Article. As follow you can find the point-to-point response to reviewer questions. In addition, the uploaded version of the manuscript contains the highlighted correction (in green).

We hope that you will find our data worth of attention for Cancers readers.

Best regards,

Sarah Allegra

Response to Reviewer 1

Manuscript titled “ Self-completion questionnaire on sleep evaluation in patients undergoing oxaliplatin therapy: an observational study. “ is a very interesting article in the field of QoL in cancer patients. The overall structure is of good quality and easy to read. Methods and Results are clear and results corroborate the initial hypothesis of the authors. Figures and Tables are of sufficient quality and easy to read as well as to understand to readers.

However, manuscript need some improvements, specifically in Introduction and/or Discussion. Here the points:

Dear Reviewer, thank you very much for the thorough review. We agree to all specific comments addressed and have revised our paper in light of the useful suggestions. Answers to the specific comments/suggestions/queries are as follows.

  1. In introduction, authors should describe not only the role of QoL in oxaliplatin-treated patients but also the incidence of cardiotoxicity and its possible correlation with sleep disturbance.

Thank you for your revision. The suggested topic has been described..

  1. No high sleep quality is related to high pro-inflammatory cytokines and NLRP3 inflammasome. Authors should describe this link and highlight on the key role of interleukin 1 beta in cancer survival and cardiovascular diseases ( cite doi: 10.26355/eurrev_202111_27124.)

Thank you for your comment. The introduction has been revised as you suggested.

Based on these changes, the article could be suitable for publication in this journal.

Special thanks for your good comments. We tried our best to improve the manuscript as your suggestion. We appreciate for your warm work earnestly, and hope that the correction will meet with approval. Thank you for the kind advice again. All authors have read and approved the re-submission of the manuscript. If you have any questions, please contact us without hesitate.

Reviewer 2 Report (Previous Reviewer 2)

Comments and Suggestions for Authors

The authors have adequately addressed my previous concerns.

Comments on the Quality of English Language

Suitable.

Author Response

Turin, February, 2024

Dear Editor,

Please, find enclosed a revised manuscript (Manuscript ID: cancers-2824919) to be considered for publication in Biomedicines, Special Issue on " Platinum-Based Therapeutics for Cancer” as Article. As follow you can find the point-to-point response to reviewer questions. In addition, the uploaded version of the manuscript contains the highlighted correction (in green).

We hope that you will find our data worth of attention for Cancers readers.

Best regards,

Sarah Allegra

Reviewer 2

The authors have adequately addressed my previous concerns.

Special thanks for your good comments. We tried our best to improve the manuscript as your suggestion. We appreciate for your warm work earnestly, and hope that the correction will meet with approval. Thank you for the kind advice again. All authors have read and approved the re-submission of the manuscript. If you have any questions, please contact us without hesitate.

Reviewer 3 Report (Previous Reviewer 3)

Comments and Suggestions for Authors

Thank you for submitting the revised version of your paper. It has been largely improved.

I still have concerns regarding the terms employed in the manuscript. Please be more specific about the fact that this paper is related to sleep complaints and not something else.

Some examples :

- Simple summary "sleep alterations" is not appropriate, this paper is about sleep complaints or subjective sleep quality and quantity (as sleep quality and quantitiy were evaluated with questionnaires)

- Abstract, aim : same

- Keywords

- Introduction : sleep-wake cycle instead of focusing on sleep

- Discussion : "sleep disorders"

Also, in the introduction, I can not see the usefulness of talking about hospital environment as this is not the purpose of the paper, I suggest to remove this section that leads to confuse information.

Finally, references should again been improved to cite papers that have been pusblished about sleep complaints in cancer patients (examples : Savard et al., 2013 ; Otte et al., 2015 ; Ratcliff et al., 2021). The review of Samuelsson et al., 2018 is also of interest.

Author Response

Turin, February, 2024

Dear Editor,

Please, find enclosed a revised manuscript (Manuscript ID: cancers-2824919) to be considered for publication in Biomedicines, Special Issue on " Platinum-Based Therapeutics for Cancer” as Article. As follow you can find the point-to-point response to reviewer questions. In addition, the uploaded version of the manuscript contains the highlighted correction (in green).

We hope that you will find our data worth of attention for Cancers readers.

Best regards,

Sarah Allegra

Reviewer 3

Thank you for submitting the revised version of your paper. It has been largely improved.

I still have concerns regarding the terms employed in the manuscript. Please be more specific about the fact that this paper is related to sleep complaints and not something else.

Some examples :

- Simple summary "sleep alterations" is not appropriate, this paper is about sleep complaints or subjective sleep quality and quantity (as sleep quality and quantitiy were evaluated with questionnaires)

- Abstract, aim : same

- Keywords

- Introduction : sleep-wake cycle instead of focusing on sleep

- Discussion : "sleep disorders"

Dear Reviewer, thank you very much for the thorough review. We agree to all specific comments addressed and have revised our paper in light of the useful suggestions.

Also, in the introduction, I can not see the usefulness of talking about hospital environment as this is not the purpose of the paper, I suggest to remove this section that leads to confuse information.

Thank you for you comment, the introduction has been revised as you suggested.

Finally, references should again been improved to cite papers that have been pusblished about sleep complaints in cancer patients (examples : Savard et al., 2013 ; Otte et al., 2015 ; Ratcliff et al., 2021). The review of Samuelsson et al., 2018 is also of interest.

Thank you for your professional revision, the papers have been citated as you suggested.

Round 2

Reviewer 3 Report (Previous Reviewer 3)

Comments and Suggestions for Authors

Thank you for having revise the manuscript, this is now fine by me.

This manuscript is a resubmission of an earlier submission. The following is a list of the peer review reports and author responses from that submission.

Round 1

Reviewer 1 Report

Comments and Suggestions for Authors

General

This is an observational retrospective questionnaire-based study.

It, however, is not about “sleep-wake cycle” or “sleep-wake rhythms”.  It is about self-reported sleep and wake “quality”.

I read the abstract, methods, and results. Based on the text in those sections, I did not review the Introduction or Discussion.

A statistician should be consulted for more appropriate and powerful statistical tests.

Specifics:

Methods line 80: Who completed the questionnaire? I think it is the patient (line 112), but am not sure.

Methods: Need CONSORT or other diagram about the number of eligible people (which can be obtained from the hospital records) vs those who competed the questionnaire.  Where there any differences between people who did or did not complete the questionnaire?

Methods: what is the relationship of the time of the data collection since cancer diagnosis? Since onset of treatment with oxaliplatin? Relative to side-effects of the treatments for the cancer?

Methods:  Not clear what the sleep variables are.

·       I think they are a single score from the Sleep Hygiene questionnaire, PSQI score, and PSQI score greater or less than 5. Is this correct

·       Lines 95-97.  I do not know what “through the evaluation of 22 behaviors, considering how many times a week in the last month have been implemented” means.

Methods Line 83: what is “qualification”. Per Table 2, it appears to be highest level of education.

Methods Analyses: statistical models that include multiple variables (including covariates from Table 1) would probably be more appropriate than single testing of each variable.

Results Table 2: How was the variable “Previous difficulty sleeping” extracted from the questionnaires? Is it the “previous difficulty sleeping” on line 92?  Was it a yes/no, categorical or continuous variable? Does it mean before the cancer or before treatment? Or, is it the previous difficultly sleeping from the PSQI (which asks about the prior month) as in Figure 3?

Results: what was the distribution of previous difficulty sleeping, sleep hygiene and PSQI scores by men and women?

Results Line 152: what is the variable “sleep quality” that is used? There is mention of “sleep hygiene” and PSQI, but not of “sleep quality” – unless it refers to PSQI scores above or below 5.

Results Figures 1 and 2: Why are only data from male patients shown? Why are the plots of liver and biliary tract cancers vs all others or pancreas vs all others if this was not an a priori hypothesis and if this is not what was specifically tested?

Results Figure 4: what were the metrics of “stress, worries, and anxiety”? Yes/no? numeric response? Categorical response? Collection of this variable is not mentioned in the Methods section.

Comments on the Quality of English Language

There are multiple errors in spelling and grammar, and text is not always clear. Review by an English-language editor is recommended.

Reviewer 2 Report

Comments and Suggestions for Authors

This study by Mussa and colleagues sought to examine the effects of oxaliplatin therapy on sleep. The authors performed this study in 144 male and female patients currently undergoing oxaliplatin therapy. Patients data were obtained using a general questionnaire, sleep hygiene questionnaire, and PSQI. Overall, there are some concerns regarding the study design and statistical comparisons described below.

Major concerns:

1.    The current study design is not appropriate to test the scientific question. There are too many confounding variables within the study. For example, the inclusion of patients taking antidepressants, benzodiazepines, antiepileptics, etc. The authors themselves describe how these drugs can alter sleep (line 278). In addition, there is no description of how many cycles of oxaliplatin therapy patients have received, or if this variable was even quantified. Where patients administered other chemotherapeutics? To examine this scientific question, appropriate exclusion criteria should be included. Currently, there are no exclusion criteria which makes the entire design difficult to draw meaningful conclusions.

 2.     Figure 1. How can one statistically compare sleep in males with a tumor in the liver or biliary track to all other tumor localizations. From the data in Table 2, there are only two males within this group. Therefore, any statistical comparison is inappropriate.

Minor

1.     Please rewrite the last two sentences of the abstract to increase clarity for the reader.

2.     The introduction is lacking references after most sentences. Please update the introduction to include appropriate references.

3.     Please updated the last sentence of the introduction: Innominato, Pasquale F., et al. "Subjective sleep and overall survival in chemotherapy-naïve patients with metastatic colorectal cancer." Sleep medicine 16.3 (2015): 391-398.

4.     Line 104 change 'was' to 'we'

Comments on the Quality of English Language

The authors should consult with a native English speaker.

Reviewer 3 Report

Comments and Suggestions for Authors

Authors submitted a paper dedicated to the effetcs of oxaliplatin on sleep-wake cycle. While the topic is of interest, I have concerns regarding this paper and its content. See below.

Globally, there is a lack of appropriate references in the introduction and in the discussion. None or almost previous studies about sleep and chemotherapy are cited. 

Also, authors indicate sleep-wake cycle in the title but starts their introduction with sleep and do not provide any definition of both concepts. What is the topic of the paper ? Sleep or sleep-wake cycle ?

The effect of sex is indicated in the title but the introduction does not provide any background about this and the effect of sex is not in the aim.

The introduction should discuss previous studies about sleep (or sleep-wake cycle) and chemotherapy in order to bring in the novelty of the current study. Also, oxaliplatin and sex are not enough introduced.

Why using a sleep hygiene questionnaire instead of a validated one ?

The methods section is not clear enough given that subtitles are missing.

I am not convinced by the validity of the statistical approach. Why using comparisons instead of regression or mixed  models ? Also, there is not correction for multiple comparisons. 

Finally, it is stated that "Performing the Kruskal–Wallis and Mann–Whitney tests to evaluate the influence of 155 all the collected variables on sleep quality" => This is not the aim given in the introduction.

Comments on the Quality of English Language

no comment